

# Can we measure beauty? Computational evaluation of coral reef aesthetics

Andreas F. Haas[1], Marine Guibert[2], Anja Foerschner[3], Tim Co[4], Sandi Calhoun[1], Emma George[1], Mark Hatay[1], Elizabeth Dinsdale[1], Stuart A. Sandin[5], Jennifer E. Smith[5], Mark J.A. Vermeij[6,7], Ben Felts[4], Phillip Dustan[8], Peter Salamon[4] and Forest Rohwer[1]

[1] Department of Biology, San Diego State University, San Diego, CA, United States
[2] ENSTA-ParisTech, Université de Paris-Saclay, Palaiseau, France
[3] The Getty Research Institute, Getty Center, Los Angeles, CA, United States
[4] Department of Mathematics and Statistics, San Diego State University, San Diego, CA, United States
[5] Scripps Institution of Oceanography, University of California San Diego, San Diego, CA, United States
[6] Caribbean Research and Management of Biodiversity (CARMABI), Willemstad, Curacao
[7] Aquatic Microbiology, University of Amsterdam, Amsterdam, Netherlands
[8] Department of Biology, College of Charleston, Charleston, SC, United States

Corresponding author
Andreas F. Haas,
andreas.florian.haas@gmail.com

## ABSTRACT

The natural beauty of coral reefs attracts millions of tourists worldwide resulting in substantial revenues for the adjoining economies. Although their visual appearance is a pivotal factor attracting humans to coral reefs current monitoring protocols exclusively target biogeochemical parameters, neglecting changes in their aesthetic appearance. Here we introduce a standardized computational approach to assess coral reef environments based on 109 visual features designed to evaluate the aesthetic appearance of art. The main feature groups include color intensity and diversity of the image, relative size, color, and distribution of discernable objects within the image, and texture. Specific coral reef aesthetic values combining all 109 features were calibrated against an established biogeochemical assessment (NCEAS) using machine learning algorithms. These values were generated for ~2,100 random photographic images collected from 9 coral reef locations exposed to varying levels of anthropogenic influence across 2 ocean systems. Aesthetic values proved accurate predictors of the NCEAS scores (root mean square error < 5 for $N \geq 3$) and significantly correlated to microbial abundance at each site. This shows that mathematical approaches designed to assess the aesthetic appearance of photographic images can be used as an inexpensive monitoring tool for coral reef ecosystems. It further suggests that human perception of aesthetics is not purely subjective but influenced by inherent reactions towards measurable visual cues. By quantifying aesthetic features of coral reef systems this method provides a cost efficient monitoring tool that targets one of the most important socioeconomic values of coral reefs directly tied to revenue for its local population.

## INTRODUCTION

Together with fishing, cargo shipping, and mining of natural resources, tourism is one of the main economic values to inhabitants of coastal areas. Tourism is one of the world's largest businesses (*Miller & Auyong, 1991*) and with ecotourism as the fastest growing form of it worldwide (*Hawkins & Lamoureux, 2001*) the industry is increasingly dependent on the presence of healthy looking marine ecosystems (*Peterson & Lubchenco, 1997*). In this context coral reefs are one of the most valuable coastal ecosystems. They attract millions of visitors each year through their display of biodiversity, their abundance of colors, and their sheer beauty and lie at the foundation of the growing tourism based economies of many small island developing states (*Neto, 2003*; *Cesar, Burke & Pet-Soede, 2003*).

Over the past decades the problem of coral reef degradation as a result of direct and indirect anthropogenic influences has been rigorously quantified (*Pandolfi et al., 2003*). This degradation affects not only the water quality, but also the abundance and diversity of the reefs inhabitants, like colorful reef fish and scleractinian corals. To assess the status of reef communities and to monitor changes in their composition through time, a multitude of monitoring programs have been established, assessing biophysical parameters such as temperature, water quality, benthic cover, and fish community composition (e.g., *Jokiel et al., 2004*; *Halpern et al., 2008*; *Kaufman et al., 2011*). These surveys however target exclusively on provisioning, habitat, and regulating ecosystem services and neglect their cultural services; i.e., the immediately nonmaterial benefits people gain from ecosystems (*Seppelt et al., 2011*; *Martin-Lopez et al., 2012*; *Casalegno et al., 2013*). Monitoring protocols to assess the biogeochemical parameters of an ecosystem, which need to be conducted by trained specialists to provide reliable data, will not give account of one of the most valuable properties of coastal environments: their aesthetic appearance to humans, which is likely the main factor prompting millions of tourists each year to visit these environments.

The value of human aesthetic appreciation for ecosystems has been studied in some terrestrial (e.g., *Hoffman & Palmer, 1996*; *Van den Berg, Vlek & Coeterier, 1998*; *Sheppard, 2004*; *Beza, 2010*; *De Pinho et al., 2014*) and marine ecosystems (*Fenton & Syme, 1989*; *Fenton, Young & Johnson, 1998*; *Dinsdale & Fenton, 2006*). However most of these studies have relied on surveys, collecting individual opinions on the aesthetic appearance of specific animals or landscapes and are therefore hard to reproduce in other locations due to a lack of objective and generalizable assessments of aesthetic properties. A recent approach by *Casalegno et al. (2013)* objectively measures the perceived aesthetic value of ecosystems by quantifying geo-tagged digital photographs uploaded to social media resources.

Although relatively new in the context of ecosystem evaluation, efforts to define universally valid criteria for aesthetic principles have been existing since antiquity (e.g., Plato, Aristotle, Confucius, Laozi). Alexander Gottlieb Baumgarten introduced aesthetics in 1735 as a philosophical discipline in his *Meditationes* (*Baumgarten & Baumgarten, 1735*) and defined it as the science of sensual cognition. Classicist philosophers such as Immanuel Kant, Georg Wilhelm Friedrich Hegel, or Friedrich Schiller, then established further theories of the "aesthetic," coining its meaning as a sense of beauty and connecting it to the visual arts. *Kant (1790)* also classified judgments about aesthetic values as having a

subjective generality. In the 20th and 21st century, when beauty was not necessarily the primary sign of quality of an artwork anymore, definitions of aesthetics and attempts to quantify aesthetic values have reemerged as a topic of interest for philosophers, art historians, and mathematicians alike (e.g., *Datta et al., 2006*; *Onians, 2007*).

With the term aesthetics recipients usually characterize the beauty and pleasantness of a given object (*Dutton, 2006*). There are however various ways in which aesthetics is defined by different people as focus of interest and aesthetic values may change depending on previous attainment (*Datta et al., 2006*). For example, while some people may simply judge an image by the pleasantness to the eye, another artist or professional photographer may be looking at the composition of the object, the use of colors and light, or potential additional meanings conveyed by the motive (*Datta et al., 2006*). Thus assessing the aesthetic visual quality of paintings seems, at first, to pose a highly subjective task (*Li & Chen, 2009*). Contrary to these assumptions, various studies successfully applied mathematical approaches to determine the aesthetic values of artworks such as sculptures, paintings, or photographic images (*Datta et al., 2006*; *Li & Chen, 2009*; *Ke, Tang & Jing, 2006*). The methods used are based on the fact that certain objects or certain features in them have higher aesthetic quality than others (*Datta et al., 2006*; *Li & Chen, 2009*). The overarching consensus hereby is that objects, or images, which are pleasing to the eye, are considered to be of higher value in terms of their aesthetic beauty. The studies which inspired the metrics used in our current work successfully extracted distinct features based on the intuition that they can discriminate between aesthetically pleasing and displeasing images. By constructing high level semantic features for quality assessment these studies have established a significant correlation between various computational properties of photographic images and their aesthetics perceptions by humans (*Datta et al., 2006*; *Li & Chen, 2009*).

## METHODS

*Study sites:* Four atolls across a gradient of human impact served as basis for this study. The 4 islands are part of the northern Line Islands group located in the central Pacific. The most northern atoll Kingman has no population and is, together with Palmyra which is exposed to sparse human impact, part of the US national refuge system. The remaining two atolls Tabuaeran and Kiritimati are inhabited and part of the Republic of Kiribati (*Dinsdale et al., 2008*; *Sandin et al., 2008*). To extend the validity of the method proposed here to other island chains and ocean systems we included an additional sampling site in the Central Pacific (Ant Atoll) and four locations in the Caribbean also subjected to different levels of human impact (2 sites on Curacao, Klein Curacao, and Barbuda, Fig. 1). From every location we collected sets of $172 \pm 17$ benthic photo-quadrant (*Preskitt, Vroom & Smith, 2004*) and $63 \pm 9$ random pictures. To evaluate the level of human impact and status of the ecosystem we used the cumulative global human impact map generated by the National Center for Ecological Analysis and Synthesis (NCEAS; http://www.nceas.ucsb.edu/globalmarine/impacts). These scores incorporate data related to: artisanal fishing; demersal destructive fishing; demersal non-destructive, high-bycatch fishing; demersal

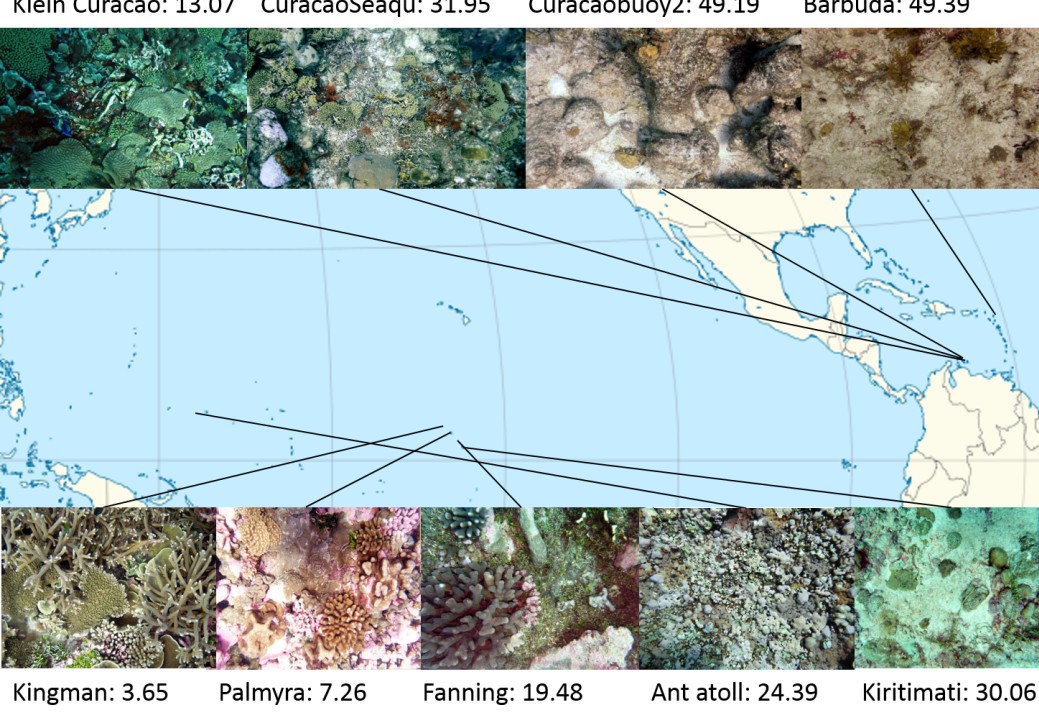

Klein Curacao: 13.07    CuracaoSeaqu: 31.95    Curacaobuoy2: 49.19    Barbuda: 49.39

Kingman: 3.65    Palmyra: 7.26    Fanning: 19.48    Ant atoll: 24.39    Kiritimati: 30.06

**Figure 1 Map of sampling sites with representative images and NCEAS scores.** The upper 4 images show images of the benthic community in the respective Caribbean sites, the lower images represent the sampling sites throughout the tropical Pacific.

non-destructive low-bycatch fishing; inorganic pollution; invasive species; nutrient input; ocean acidification; benthic structures; organic pollution; pelagic high-bycatch fishing; pelagic low-bycatch fishing; population pressure; commercial activity; and anomalies in sea surface temperature and ultraviolet insolation (*Halpern et al., 2008*; *McDole et al., 2012*). Additionally, bacterial cell abundance across the 4 Northern Line Islands and 3 of the Caribbean locations (Curacao main island and Barbuda; Table 1) were measured after the method described by *Haas et al. (2014)*.

*Aesthetic feature extraction:* In total we extracted, modified, and complemented 109 features (denoted as $f_1, f_2, \ldots, f_{109}$) from three of the most comprehensive studies on computational approaches to aesthetically evaluate paintings and pictures (Table S1; *Datta et al., 2006*; *Li & Chen, 2009*; *Ke, Tang & Jing, 2006*). Aesthetic evaluation of paintings and photographs in all three studies were based on surveys of randomly selected peer groups. Some of the features presented in previous work were however difficult to reproduce owing to insufficient information given on these features (e.g., *f16–24*, or *f51*). This may have led to slight alterations in some of the codes which were inspired by the suggested features but deviate slightly in their final form. As the pictures were considered to be objective samples representing the respective seascape, some traditional aesthetic features, like size of an image or its aspect ratio have not been considered in this study. Overarching feature groups considered in the picture analysis were color, texture, regularity of shapes, and relative sizes and positions of objects in each picture.

*Aesthetic value:* Although some of the implemented codes appeared similar and were assessing closely related visual aspects, all of the suggested codes were implemented and their value, or potential redundancy, was evaluated using machine learning algorithms. Following feature extraction the 109 feature values were used as input for feed forward neural networks that optimize the importance of features or feature groups and generate a single aesthetic value for each respective photograph. The target outputs for the training of the networks were the NCEAS scores of the regions where the pictures were taken. The pictures were randomly separated into a batch used for training the machine learning algorithms ($N = 1,897$) and one on which the algorithms were tested ($N = 220$, 20 from each of 11 sites). We used Matlab's neural network package on the training samples which further subdivided these samples into training (70%), validation (15%) and test (15%) sets (see Appendix for details). Unlike previous studies in which the aesthetic quality was classified in given categories, this machine learning regression approach generates a continuous metric for the aesthetic quality of a given reefscape.

## RESULTS

An aesthetic value of coral reef images was defined using features previously created for measuring the aesthetic quality of images. The values were calibrated using machine learning to match NCEAS scores as closely as possible. Our algorithm gleaned the NCEAS score from an image to within a root mean squared (rms) error of 6.57. Using five images from the same locale improved the NCEAS score prediction to an rms error of 4.46. The relative importance for each feature derived from a random forests approach showed that all three overarching feature groups, texture, color of the whole image, and the size, color, and distribution of objects within an image yielded important information for the algorithm (Fig. S1). The ten most important features, or feature groups were hereby the similarity in spatial distribution of high frequency edges, the wavelet features, number of color based cluster, the area of bounding boxes containing a given percentage of the edge energy, the average value of the HSV color space, entropy of the blue matrix, range of texture, the arithmetic and the logarithmic average of brightness, and the brightness of the focus region as defined by the rule of thirds.

The mean coral reef aesthetic values generated with this approach for each picture were significantly different ($p < 0.001$) between all sampling locations except for Ant Atoll, Fanning and Klein Curacao (ANOVA followed by Tukey, see Table S2). These sites are also exposed to comparable levels of anthropogenic disturbance (NCEAS: 14.11–19.48). Regression of coral reef aesthetic values against the NCEAS scores of the respective sampling site showed a significant correlation ($p < 0.001$) for both the training ($n = 1,897$, $R^2 = 0.93$) and the test ($n = 220$, $R^2 = 0.80$) set of images (Fig. 2). Further comparison to microbial abundance, available for 7 of the 9 locations (microbial numbers for Curacao Buoy2 and Ant Atoll were not available), revealed a significant correlation between the aesthetic appearance of the sampling sites and their microbial density ($p = 0.0006$, $R^2 = 0.88$; Fig. 3).

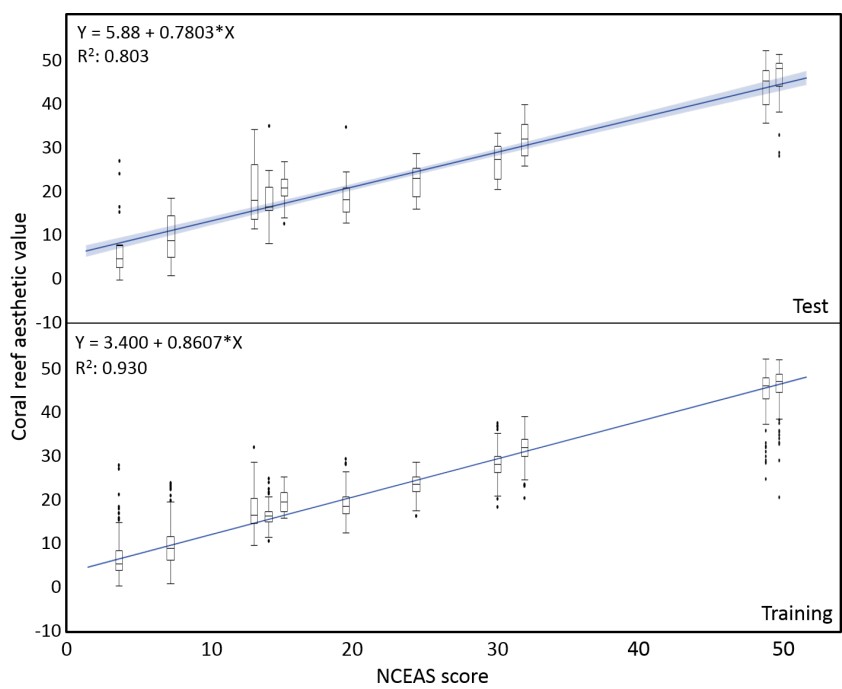

**Figure 2 Coral reef aesthetic values.** Boxplots of coral reef aesthetic values at each site and regression of coral reef aesthetic values vs. NCEAS scores across all assessed reef sites. (Test) shows coral reef aesthetic values calculated for 200 images on which the previously trained machine learning algorithm was tested. (Training) shows the generated coral reef aesthetic values from 1,970 images used to improve the feed forward neural networks that optimized the importance of features or feature groups in generating a single coral reef aesthetic value.

## DISCUSSION

This is the first study using standardized computational approaches to establish a site-specific correlation between aesthetic value, ecosystem degradation, and the microbialization (*McDole et al., 2012*) of marine coral reef environments.

### Human response to visual cues

The connection between reef degradation and loss of aesthetic value for humans seems intuitive but initially hard to capture with objective mathematical approaches. *Dinsdale (2009)* showed that human visual evaluations provided consistent judgment of coral reef status regardless of their previous knowledge or exposure to these particular ecosystems. The most important cue was the perceived health status of the system. Crucial for this intuitive human response to degraded or "unhealthy" ecosystems is the fact that we are looking at organic organisms and react to them with the biological innate emotion of disgust (*Curtis, 2007*; *Hertz, 2012*). Being disgusted is a genetically anchored reaction to an object or situation, which might be potentially harmful to our system. Often, a lack of salubriousness of an object or situation is the crucial element for our senses, one of them visual perception, to signal us to avoid an object or withdraw from a situation (*Foerschner, 2011*). As the microbial density and the abundance of potential pathogens in degrading reefs are significantly elevated (*Dinsdale et al., 2008*)—albeit not visible

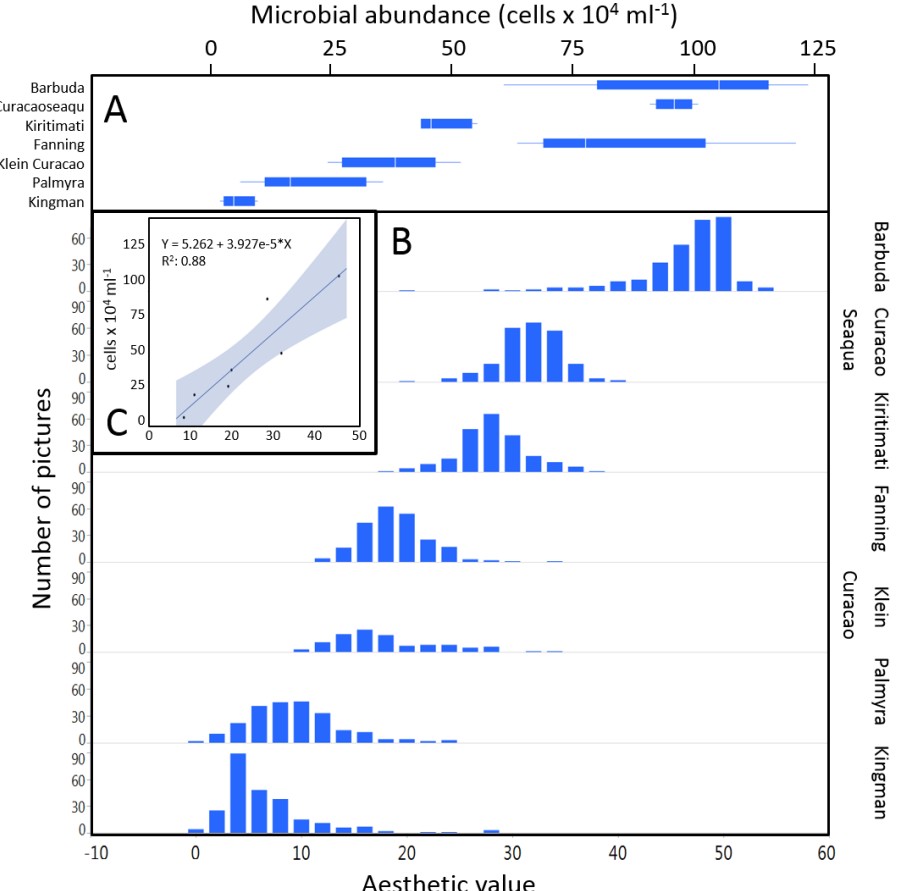

**Figure 3 Distribution of aesthetic values.** (A) shows microbial cell abundance at 7 reef sites. (B) shows the distribution of pictures with respective aesthetic values at each of those sites. (C) shows the regression between mean microbial cell abundance and mean aesthetic value (training + test) across all 7 sampling sites.

to the human eye—our inherent human evaluation of degraded reefs as aesthetically unpleasing, or even disgusting, is nothing else than recognizing the visual effects of these changes as a potential threat for our well-being. Generally the emotion of disgust protects the boundaries of the human body and prevents potentially harmful substances from compromising the body. This theory was supported by French physiologist *Richet (1984)*, who described disgust as an involuntary and hereditary emotion for self-protection. The recognition of something disgusting, and thus of a lack of aesthetic value, prompts an intuitive withdrawal from the situation or from the environment triggering this emotion. Recent evolutionary psychology largely follows this thesis and concludes that disgust, even though highly determined by a certain social and cultural environment, is genetically imprinted and triggered on a biological level by objects or environments which are unhealthy, infectious, or pose a risk to the human wellbeing (*Rozin & Fallon, 1987*; *Rozin & Schull, 1988*; *Foerschner, 2011*). Decisive here is the connection between disgust and the salubriousness, or better lack thereof, of given objects, which indicates unhealthiness. Our here presented study supports these theories by establishing objectively quantifiable coral

reef aesthetic values for ecosystems along a gradient of reef degradation, and for a subset microbial abundance. Perception of aesthetic properties is not purely a subjective task and measurable features of aesthetic perception are inherent to human nature. The main visual features assessed by our analysis are color intensity and diversity, relative size and distribution of discernable objects, and texture (Fig. S1 and Table S1). Human perception of each of these features does not only trigger innate emotions, each of these features also yields palpable information on the status of the respective ecosystem.

*Color:* Thriving ecosystems are abounding with bright colors. On land photosynthesizing plants display a lush green and, at least seasonally, blossoms and fruits in every color. Animals display color for various reasons, for protective and aggressive resemblance, protective and aggressive mimicry, warning colors, and colors displayed in courtship (*Cope, 1890*). Underwater, coral reefs surpass all other ecosystems in their display of color. The diversity and colorfulness of fauna and flora living in healthy reef systems is unmatched on this planet (*Marshall, 2000*; *Kaufman, 2005*). This diverse and intense display of color is, however, not only an indicator of high biodiversity, but also of a "clean" system. The brightest and most diverse display of colors by its inhabitants will be dampened in a system with foggy air or murky waters. Previous studies suggest an evolutionary theory in the human preference of color patterns as a result of behavioral adaptations. *Hurlbert & Ling (2007)* conclude that color preferences are engrained into human perception as neural response to selection processes improving performance on evolutionarily important behavioral tasks. Humans were more likely to survive and reproduce successfully if they recognize objects or environments that characteristically have colors which are advantageous/disadvantageous to the organism's survival, reproductive success, and general well-being (*Palmer & Schloss, 2010*). Thus it is again not surprising that humans are inherently drawn to places with bright and diverse colors as they represent clean systems not associated with pollution or other health risks.

*Objects:* Not only does the visual brain recognize properties like luminance or color, it also segregates higher-order objects (*Chatterjee, 2014*). The relative size, distribution and regularity of objects in the pictures analyzed were important features in determining the aesthetic value of pictures. *Birkhoff (1933)* proposed in his theory of preference for abstract polygon shapes that aesthetic preference varies directly related to the number of elements. Further it has been established that people tend to prefer round regular and convex shapes as they are more symmetrical and structured (*Jacobsen & Höfel, 2002*; *Palmer & Griscom, 2013*). The fluency theory provides an additional explanation for a general aesthetic preference for specific objects (*Reber, Winkielman & Schwarz, 1998*; *Reber, Schwarz & Winkielman, 2004*; *Reber, 2012*). It predicts aesthetic inclination as a result of many low-level features (*Oppenheimer & Frank, 2008*), like preferences for larger (*Silvera, Josephs & Giesler, 2002*), more symmetrical (*Jacobsen & Höfel, 2002*), more contrastive objects (*Reber, Winkielman & Schwarz, 1998*; reviewed in *Reber, Schwarz & Winkielman, 2004*). From a biological view there may be additional causes for the preference of larger discernable objects. Bigger objects representing living entities indicate that the environment is suitable for large animals and can provide a livelihood for apex predators

like humans, while small objects suggest a heavily disturbed system, unable to offer resources for growth or a long life experience for its inhabitants. The lack of discernable objects like fish, hard corals, or giant clams suggests that microbiota are dominant in this system, likely at the expense of the macrobes (*McDole et al., 2012*).

*Texture:* Another important criterion in the aesthetic evaluation of an image is the existence of clearly discernible outlines; a distinguishable boundary texture that keeps objects separated from their environment. The Russian philosopher *Bakhtin (1941)* elevated this characteristic to be the main attribute of grotesqueness in relation to animated bodies. Anything that disrupts the outline, all orifices or products of inner, bodily processes such as mucus, saliva, or semen evokes a negative emotional response of disgust and repulsion (*Foerschner, 2011*; *Foerschner, 2013*). Even though various theories on triggers for disgust exist, the absence of discernable boundaries (both physical and psychological) are fundamental to all of them (*Foerschner, 2011*; *Menninghaus, 2012*). For living organisms the transgression of boundaries and the dissolution of a discernable surface texture signify much more than the mere loss of form: it comprehends the organism in a state of becoming and passing, ultimately in its mortality. Decomposition, disease, and decay are as disgusting to us as mucus, saliva, or slime; the former in their direct relation to death, the latter ones as products of bodily functions, which equally identifies our organic state as transient (*Kolnai, 2004*). Further, amorphous slime covering and obscuring the underlying texture of objects may be the result of biofilm formation. A biofilm is a group of microorganisms which, frequently embedded within a mucoidal matrix, adheres to various surfaces. These microbial assemblages are involved in a wide variety of microbial infections (*Costerton, Stewart & Greenberg, 1999*). Human perception is therefore more likely to evaluate a viscous, slimy, or amorphous object surface as repulsive whereas surface textures with clearly defined boundaries and patterns are pleasing to our senses and generally deemed aesthetic.

It has to be mentioned that by no means do we claim to provide an assessment for the value of art or artistic images by this method. The value of an artwork depends not only on the aesthetics, but also on the social, economic, political or other meanings it conveys (*Adorno, 1997*), and on the emotions and impulses it triggers in a recipient. However this study suggests that perception of aesthetic properties may be more objective than commonly appraised and patterns of aesthetic evaluation are inherent to human perception.

## Crowd sourcing & historic data mining

The approach provided here will likely be a valuable tool to generate assessments on the status of reef ecosystems, unbiased by the respective data originator. By taking a set of random photographic images from a given system information on the aesthetic value and thus on the status of the ecosystem can be generated. Contrary to all previously introduced monitoring protocols the objective analysis of pictures will overcome bias introduced by the individual taking samples or analyzing the respective data. Obviously, the analysis of a single picture will depend on the motive chosen or camera handling and not every single picture will accurately reflect the status of the ecosystem (Fig. 4). However, as in most ecological approaches the accuracy of the information increases with

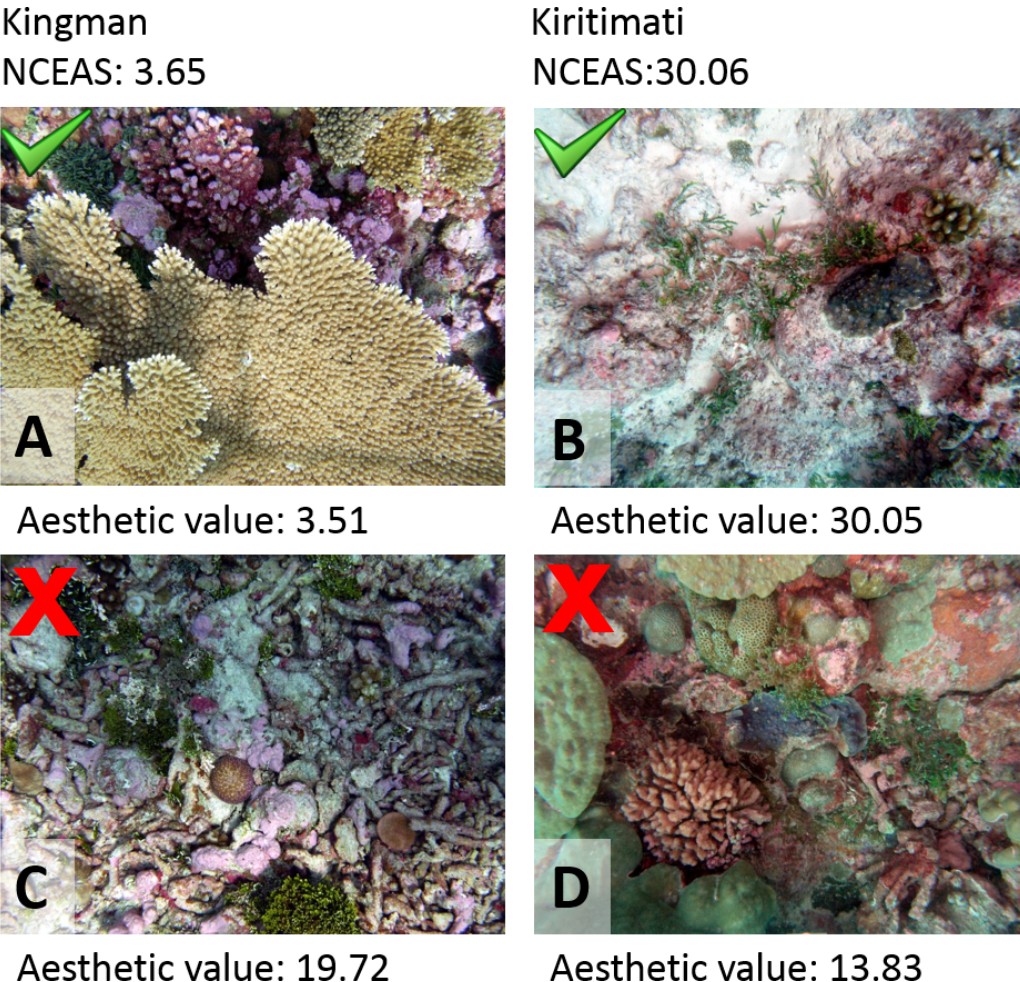

Figure 4 **Image examples.** Examples of pictures with their respective generated aesthetic values from two contrasting sampling sites, Kingman and Kiritimati. Aesthetic values for (A) and (B), which resemble representative images of the specific locations, were close matches to the NCEAS score at the respective site. (C) and (D) give examples of pictures which resulted in mismatches to the respective NCEAS score.

sample size, i.e., number of digital images available (see Fig. 3B). The application of this method to resources like geo-tagged digital image databases or historic images of known spatial and temporal origin will allow access to an immense number of samples and could provide objective information on the status and the trajectories of reefs around the world. Previous studies already focused on the problem of establishing a baseline for pristine marine ecosystems, especially coral reefs. But coral reefs are among the most severely impacted systems on our planet (*Knowlton, 2001*; *Wilkinson, 2004*; *Bellwood et al., 2004*; *Pandolfi et al., 2005*; *Hoegh-Guldberg et al., 2007*) and most of the world's tropical coastal environments are so heavily degraded that pristine reefs are essentially gone (*Jackson et al., 2001*; *Knowlton & Jackson, 2008*). The here presented method could provide a tool to establish a global baseline of coral reef environments, dating back to the first photographic coverage of the respective reef systems. As an example we used photographic images of the

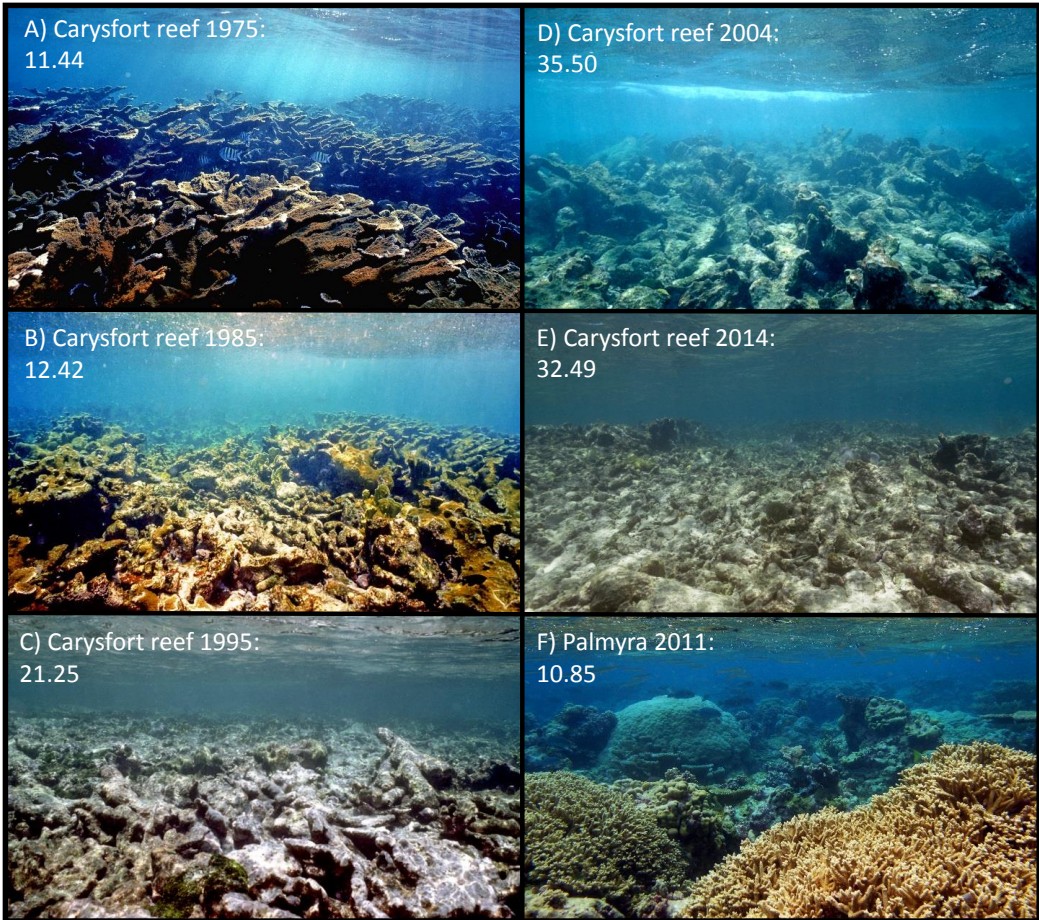

**Figure 5 Aesthetic values of Carysfort reef.** (A–E) are taken at the identical location on Carysfort reef, US Caribbean, over a time span of 40 years (photos taken by P Dustan). The aesthetic value calculated for each picture shows a significant degradation of aesthetic appearance during this period. The historic images from 1975 indicate that the aesthetic appearance of this Caribbean reef was comparable to present day pristine reefscapes as for example on Palmyra atoll in the Central Pacific (F, photo taken by J Smith).

Carysfort reef in the US Caribbean, taken at the same location over a time span of nearly 40 years (1975–2014). The image analysis showed a clear degradation of aesthetic values during those four decades (Fig. 5). While the aesthetic appearance of this Caribbean reef in 1975 is comparable to reefscapes as they are found on remote places like the Palmyra atoll today, the aesthetic value drastically declined over the 40 year time span and place the aesthetic appearance of this reef below the heavily degraded reef sites of Kiritimati today (2004 and 2014).

## Socioeconomic assessment for stakeholders

This study provides an innovative method to objectively assess parameters associated with a general aesthetic perception of marine environments. Although converting the aesthetic appearance of an entire ecosystem in simple numbers will likely evoke discussions and in some cases resentment, it may provide a powerful tool to disclose effects of implementing

conservation measurements on the touristic attractiveness of coastal environments to stakeholders. The approach allows for a rapid analysis of a large number of samples and thus provides a method to cover ecosystems on large scale. Linking aesthetic values to cultural benefits and ultimately revenue for the entire community may be an incentive to further establish and implement protection measurements and could help to evaluate the success and the value to the community of existing conservation efforts. Using monitoring cues that directly address inherent human emotions will more likely motivate and sustain changes in attitude and behavior towards a more sustainable usage of the environmental resources than technical terms and data that carry no local meaning (*Carr, 2002*; *Dinsdale, 2009*). Quantifying the aesthetic appearance of these ecosystems targets on one of the most important socioeconomic values of these ecosystems, which are directly tied to culture and the revenue of its local population.

## ACKNOWLEDGEMENTS

We thank the biosphere foundation for providing the pictures of Carysfort reef. We further thank the Captain, Martin Graser, and crew of the M/Y Hanse Explorer.

## APPENDIX: FEATURE EXTRACTION

### Global features

Global features are computed over all the pixels of an entire image.

*Color:* The HSL (hue, saturation, lightness) and HSV (hue, saturation, value) color spaces are the two most common cylindrical-coordinate representations of points in an RGB color model. The HSV and HSL color space define pixel color by its hue, saturation and value, respectively lightness (*Joblove & Greenberg, 1978*). This provides a color definition similar to the human visual perception. The first step for each picture analysis was therefore to calculate the average hue, saturation and value respectively lightness for both color spaces. Assuming a constant hue, the definition of saturation and of value and lightness are very much different. Therefore hue, saturation, and value of a pixel in the HSV space will be denoted as $I_H(m,n)$, $I_S(m,n)$ and $I_V(m,n)$, and hue, saturation and lightness in the HSL space as $I_{H\_}(m,n)$, $I_{S\_}(m,n)$ and $I_{L\_}(m,n)$ from here on, where $m$ and $n$ are the number of rows and columns in each image.

$$f_1 = \frac{1}{MN} \sum_n \sum_m I_H(m,n) \tag{1}$$

$$f_2 = \frac{1}{MN} \sum_n \sum_m I_S(m,n) \tag{2}$$

$$f_3 = \frac{1}{MN} \sum_n \sum_m I_V(m,n) \tag{3}$$

$$f_4 = \frac{1}{MN} \sum_n \sum_m I_{S\_}(m,n) \tag{4}$$

$$f_5 = \frac{1}{MN} \sum_n \sum_m I_{L\_}(m,n). \tag{5}$$

To assess colorfulness the RGB color space was separated in 64 cubes of identical volume by dividing each axis in four equal parts. Each cube was then considered as individual sample point and color distribution $D_1$ of each image defined as the frequency of color occurrence within each of the 64 cubes. Additionally a reference distribution $D_0$ was generated so that each sample point had a frequency of 1/64. The colorfulness of an image was then defined as distance between these two distributions, using the Quadratic-form distance (*Ke, Tang & Jing, 2006*) and the Earth Mover's Distance (EMD). Both features take the pair-wise euclidian distances between the sample points into account. Assuming $c_i$ is now the center position of the $i$-th cube, we get $d_{ij} = \|rgb2luv(c_i) - rgb2luv(c_j)\|_2$ after a conversion to the LUV (Adams chromatic valence space; *Adams, 1943*) color space. This leads to

$$f_6 = \sqrt{\left(h - h^0\right) T A \left(h - h^0\right)} \quad \text{and} \quad f_7 = emd(D_1, D_0, \{d_{ij}|1 < i, j < 64\}) \tag{6}$$

in which $h$ and $h_0$ are vectors listing the frequencies of color occurrence in $D_1$ and $D_0$. $A = (a_{ij})$ is a similarity matrix with $a_{ij} = 1 - d_{ij}/d_{max}$ and $d_{max} = \max(d_{ij})$; '*emd*' denotes the earth mover's distance we implemented using an algorithm described by *Rubner, Tomasi & Guibas (2000)*.

For color analysis only pixels with a saturation $I_{s\_}(m, n) < 0.2$ and a lightness $I_{L\_} \in [0.15, 0.95]$ were used as the human eye is unable to distinguish hues and only sees shades of grey outside this range. As $P_H = \{(m', n')|I_{S\_} > 0.2 \text{ and } 0.15 < I_{L\_} < 0.95\}$ represents the set of pixels whose hues can be perceived by humans, $f_8$ was defined as the most frequent hue in each image and $f_9$ as the standard deviation of colorfulness.

$$f_8 = \min(h_{max}), \tag{7}$$

where $\forall$ hue $h$, # of $\{(m', n') \in P_H|I_{H\_} = h_{max}\} \geq$ # of $\{(m', n')\} \in P_H|I_{H\_} = h$. If hues had an identical cardinal, the smallest one was chosen.

$$f_9 = \text{std}(\text{var}(I'_{H\_})). \tag{8}$$

where $I'_{H\_}(m, n) = I_{H\_}(m, n)$ if $(m, n) \in P_H$; otherwise $I'_{H\_}(m, n) = 0$. var $(I'_{H\_})$ is the vector containing the variance of each column of $I'_{H\_}$, and std returns its standard deviation.

The hue interval [0, 360] was then uniformly divided into 20 bins of identical size and computed into a hue histogram of the image. $Q$ represents the maximum value this histogram and the hue count was defined as the number of bins containing values greater than $C \cdot Q$. The number of missing hues represents bins with values smaller than $c \cdot Q$. $C$ and $c$ was set to 0.1 and 0.01, respectively.

$$f_{10} = \# \text{ of } \{i|h(i) > C \cdot Q\} \tag{9}$$
$$f_{11} = \# \text{ of } \{i|h(i) < c \cdot Q\}. \tag{10}$$

Hue contrast and missing hues contrast was computed as:

$$f_{12} = \max(\|c_h(i) - c_h(j)\|_{al}) \quad \text{with } i, j \in \{i|h(i) > C \cdot Q\} \tag{11}$$
$$f_{13} = \max(\|c_h(i) - c_h(j)\|_{al}) \quad \text{with } i, j \in \{i|h(i) < c \cdot Q\} \tag{12}$$

where $c_h(i)$ is the center hue of the $i$-th bin of the histogram and $\|\cdot\|_{al}$ refers to the arc-length distance on the hue wheel. $f_{14}$ denotes the percentage of pixels belonging to the most frequent hue:

$$f_{14} = Q/N \quad \text{where } N = \# \text{ of } P_H \tag{13}$$

$$f_{15} = 20 - \# \text{ of } \{i | h(i) > C_2 \cdot Q\} \quad \text{with } C_2 = 0.05 \tag{14}$$

*Color models:* As some color combinations are more pleasant for the human eye than others (*Li & Chen, 2009*), each image was fit against one of 9 color models (Fig. S2K). As the models can rotate, the $k$-th model rotated with an angle $\alpha$ as $M_k(\alpha)$, $G_k(I_{H\_}(m,n)$ was assigned to the grey part of the respective model. $E_{M_k}(\alpha)(m,n)$ was defined as the hue of $G_k(\alpha)$ closest to $I_{H\_}$.

$$E_{M_k}(\alpha)(m,n) = \begin{cases} I_{H\_}(m,n) & \text{if} \quad I_{H\_}(m,n) \in G_k(\alpha) \\ H_{\text{nearestborder}} & \text{if} \quad I_{H\_}(m,n) \notin G_k(\alpha) \end{cases} \tag{15}$$

where $H_{\text{nearestborder}}$ is the hue of the sector border in $M_k(\alpha)$ closest to the hue of pixel $(m,n)$. Now the distance between the image and the model $M_k(\alpha)$ can be computed as

$$F_{k,\alpha} = \frac{1}{\sum_m \sum_m I_{S\_}(m,n)} \sum_n \sum_m \|E_{Mk(\alpha)}(m,n) - I_{H\_}(m,n)\|al \cdot I_{S\_}(m,n) \tag{16}$$

with $I_{S\_}(m,n)$ accounting for less color differences with lower saturation. This definition of the distance to a model was inspired by *Datta et al. (2006)* with the addition of a normalization $\frac{1}{\sum_m \sum_n I_{s\_}(m,n)}$ which allows for a comparison of different sized images. As the distances of an image to each model yield more information than the identity of the single model the image fits best, all distances were calculated and features $f_{16}$–$f_{24}$ are therefore defined as the smallest distance to each model:

$$f_{15+k} = \min_\alpha F_{k,\alpha}, \quad k \in \{1,\ldots,9\}. \tag{17}$$

Theoretically the best fitting hue model could be defined as $M_{ko}(\alpha_o)$ with

$$\alpha(k) = \arg\min_\alpha F_{k,\alpha}, k_0 = \arg\min_{k\in\{1,\ldots9\}} F_{k,\alpha(k)} \quad \text{and} \quad \alpha_0 = \alpha(k_0). \tag{18}$$

Those models are, however, very difficult to fit. Therefore we set a threshold *TH* assuming that if $F_{k,\alpha(k)} < TH$, the picture fits the $k$-th color model. If $\forall k\, F_{k,\alpha(k)} \geq TH$ the picture was fit to the closest model. In case several models could be assigned to an image not the closest one, but the most restrictive was chosen. As the color models are already ordered according to their restrictiveness the fit to the color model we characterize as:

$$f_{25} = \begin{cases} \max k \in \{j | F_j, \alpha(j), TH\}\, k & \text{if} \quad \exists k \in \{1,\ldots,9\}, F_{k,\alpha(k)} < TH \\ k0 & \text{if} \quad \forall k\, F_{k,\alpha(k)} \geq TH \end{cases} \tag{19}$$

Normalizing the distances to the models enabled us to set a unique threshold ($TH = 10$) for all the images independently of their size.

*Brightness:* Light conditions captured by a given picture are some of the most noticeable features involved in human aesthetic perception. Some information about the light condition is already explored by the previously described color analysis, however, analyzing the brightness provides an even more direct approach to evaluating the light conditions of a given image. There are several ways to measure the brightness of an image. For this study, we implemented analysis which target slightly different brightness contrasts.

$$f_{26} = \frac{1}{MN} \sum_m \sum_n L(m,n) \tag{20}$$

$$f_{27} = \exp\left( \frac{255}{MN} \sum_m \sum_n \log\left( \in + \frac{L(m,n)}{255} \right) \right) \tag{21}$$

where $L(m;n) = (I_r(m;n) + I_g(m;n) + I_b(m;n))/3$. $f_{26}$ represents the arithmetic and $f_{27}$ the logarithmic average brightness; the latter takes the dynamic range of the brightness into account. Different images can therefore equal in one but differ in the other value. The contrast of brightness was assessed by defining $h_1$ as a histogram with 100 equally sized bins for brightness $L(m;n)$, with $d$ as index for the bin with the maximum energy $h_1(d) = \max(h_1)$. Two indices $a$ and $b$ were set as the interval $[a;b]$ which contains 98% of the energy of $h_1$. The histogram was then analyzed step by step towards both sides starting from the $d$th bin to identify $a$ and $b$. The first measure of the brightness contrast is then

$$f_{28} = b - a + 1. \tag{22}$$

For the second contrast quality feature a brightness histogram $h_2$ with 256 bins comprising the sum of the gray-level histograms $h_r, h_g$ and $h_b$ generated from the red, green and blue channels:

$$h_2(i) = h_r(i) + h_g(i) + h_b(i), \quad \forall i \in \{0,\ldots,255\}. \tag{23}$$

The contrast quality $f_{29}$ is then the width of the smallest interval $[a_2, b_2]$ where $\sum_{i=a2}^{b2} h_2(i) > 0.98 \sum_{i=0}^{255} h_2(i)$.

$$f_{29} = b_2 - a_2. \tag{24}$$

*Edge features:* Edge repartition was assessed by looking for the smallest bounding box which contains a chosen percentage of the energy of the edges, and compare its area to the area of the entire picture. Although *Li & Chen (2009)* and *Ke, Tang & Jing (2006)* offer two different versions to target this feature, both use the absolute value of the output from a $3 \times 3$ Laplacian filter with $\alpha = 0.2$. For color images the R, G and B channels are analyzed separately and the mean of the absolute values is used. At the boundaries the values outside the bounds of the matrix was considered equal to the nearest value in the matrix borders. According to *Li & Chen (2009)* the area of the smallest bounding box, containing 81% of the edge energy of their 'Laplacian image' (90% in each direction), was divided by the area

of the entire image (Figs. S2E–S2H).

$$f_{30} = H_{90}W_{90}/HW \tag{25}$$

$H_{90}$ and $W_{90}$ represent the height and width of the bounding box with $H$ and $W$ as the height and width of the image.

*Ke, Tang & Jing (2006)* resized each Laplacian image initially to $100 \times 100$ and the image sum was normalized to 1. Subsequently the area of the bounding box containing 96.04% of the edge energy (98% in each direction) was established and the quality of the image was defined as $1 - H_{98}W_{98}$, whereby $H_{98}$ and $W_{98}$ are the height and width of the bounding box.

$$f_{31} = 1 - H_{98}W_{98}; H_{98} \quad \text{and} \quad W_{98} \in [0,1]. \tag{26}$$

Resizing and normalizing the Laplacian images further allows for an easy comparison of different Laplacian images. Analog to *Ke, Tang & Jing (2006)* who compared one group of professional quality photos and one group of photos of inferior quality, we can now consider two groups of images: one with pictures of pristine and one with pictures of degraded reefs. $M_p$ and $M_s$ represent the mean Laplacian image of the pictures in each of the respective groups. This allows a comparison of the Laplacian image $L$ with $M_p$ and $M_s$ using the $L1$-distance.

$$f_{32} = d_s - d_p, \quad \text{where} \tag{27}$$

$$d_s = \sum_{m,n} |L(m,n) - M_s(m,n)| \tag{28}$$

$$d_p = \sum_{m,n} |L(m,n) - M_p(m,n)|. \tag{29}$$

The sum of edges $f_{33}$ was added as an additional feature not implemented by one of the above mentioned studies. Sobel image $S$ of a picture was defined as a binary image of identical size, with 1's assigned to edges present according to the Sobel method and 0's for no edges present. For a color image Sobel images $S_r$, $S_g$ and $S_b$ were constructed for each of its red, green and blue channels and the sum of edges defined as

$$f_{33} = (|S_r|_{L1} + |S_g|_{L1} + |S_b|_{L1})/3. \tag{30}$$

*Texture analysis:* To analyze the texture of pictures more thoroughly we implemented features not yet discussed in *Ke, Tang & Jing (2006)*, *Datta et al. (2006)*, or *Li & Chen (2009)*. Therefore we considered $R_H$ to be a matrix of the same size as $I_H$, where each pixel $(m,n)$ contains the range value (maximum value–minimum value) of the 3-by-3 neighborhood surrounding the corresponding pixel in $I_H$. $R_S$ and $R_V$ were computed in the same way for $I_S$ and $I_V$ and the *range of texture* was defined as

$$f_{34} = \frac{1}{MN} \sum_m \sum_n (R_H(m,n) + R_S(m,n) + R_V(m,n))/3. \tag{31}$$

Additionally $D_H, D_S$, and $D_V$ were set as the respective matrix identical in size to $I_H, I_S$, and $I_V$, where each pixel $(m, n)$ contains the standard deviation value of the 3-by-3 neighborhood around the corresponding pixel in $I_H, I_S$, or $I_V$. The average standard deviation of texture was defined as:

$$f_{35} = \frac{1}{MN} \sum_m \sum_n (D_H(m,n) + D_S(m,n) + D_V(m,n))/3. \tag{32}$$

The entropy of an image is a statistical measure of its randomness, and can also be used to characterize its texture. For a gray-level image, it is defined as—$\sum_{i=0}^{255} p(i) * \log_2(p(i))$ where $p$ is a vector containing the 256 bin gray-level histogram of the image. Thus, we define features $f_{36}, f_{37}$ and $f_{38}$ as the entropy of $I_r, I_g$, and $I_b$ respectively.

$$f_{36} = \text{entropy}(I_r) \tag{33}$$

$$f_{37} = \text{entropy}(I_g) \tag{34}$$

$$f_{38} = \text{entropy}(I_b). \tag{35}$$

*Wavelet based texture:* Texture feature analysis based on wavelets was conducted according to *Datta et al. (2006)*. However concrete information on some of the implemented steps (e.g., norm or exact Daubechies wavelet used) was sometimes not available which may result in a slight deviation of the calculation. First a three level wavelet transformation on $I_H$ was performed using the Haar Wavelet (see Figs. S2I and S2J). A 2D wavelet transformation of an image yields 4 matrices: the approximation coefficient matrix $C^A$ and the three details coefficient matrices $C^H, C^V$ and $C^D$. Height and width of resulting matrices are 50% of the input image and $C^H, C^V$ and $C^D$ show horizontal, vertical and diagonal details of the image. For a three-level wavelet transformation a 2D wavelet transformation is performed and repeated on the approximation coefficient matrix $C_1^A$ and repeated again on the new approximation coefficient matrix $C_2^A$, resulting in 3 sets of coefficients matrices. The $i$th-level detail coefficient matrices for the hue image $I_H$ were then denoted as $C_i^H, C_i^V$, and $C_i^D (I \in \{1, 2, 3\})$. Features $f_{39}$–$f_{41}$ are then defined as follows:

$$f_{38+i} = \frac{1}{Si} \sum_m \sum_n (C_i^H(m,n) + C_i^V(m,n) + C_i^D(m,n)), \quad i \in \{1, 2, 3\} \tag{36}$$

where $\forall i \in \{1, 2, 3\}$, $S_i = |C_i^H|_{L1} + |C_i^V|_{L1} + |C_i^D|_{L1}$. Features $f_{42}$–$f_{44}$ and $f_{45}$–$f_{47}$ recomputed accordingly for $I_s$ and $I_v$. Features $f_{48}$–$f_{50}$ are defined as the sum of the three wavelet features for $H, S$, and $V$ respectively:

$$f_{48} = \sum_{i=40}^{42} f_i, f_{49} = \sum_{i=43}^{45} f_i, f_{50} = \sum_{i=46}^{48} f_i. \tag{37}$$

*Blur:* Measurements of the image blur were done based on suggestions given by *Li & Chen (2009)* and *Ke, Tang & Jing (2006)*. Based on the information provided we were not able to implement the features successfully, thus the features presented here are a modified

adaptation. For this purpose each picture was considered to be a blurred image $I_{\text{blurred}}$ as a result of the convolution of an hypothetical sharp version of the image $I_{\text{sharp}}$ and a Gausssian filter $G_\sigma : I_{\text{blurred}} = G_\sigma * I_{\text{sharp}}$. As the Gaussian filter eliminates high frequencies only, the blur of a picture can be determined by quantifying the frequency of the image above a certain threshold $\theta$. A higher frequency indicates less blur. The threshold $\theta$ reduces the noise and provides a defined cutoff of the high frequencies. To quantify blur in a given image, a 2D Fourrier Transform was performed resulting in $Y$. To avoid ambiguities the 2D Fourrier Transform is then normalized by $1/\sqrt{MN} : Y = fft2(I_{\text{blurred}})/\sqrt{MN}$. As we observed a phenomenon of spatial aliasing, only the frequencies $(m', n')$ where $0 < m' < M/2$ and $0 < n' < N/2$ were used, resulting in

$$f_{51} = \max\left(2\frac{m' - \left[\frac{M}{2}\right]}{M} ; 2\frac{n' - \left[\frac{N}{2}\right]}{N}\right) \tag{38}$$

where $|Y(m', n')| > \theta$, $0 < m' < M/2$, and $0 < n' < N/2$. The threshold was set as $\theta = 0.45$.

## Local features

In addition to global features which provide information about the general aspect of a picture, local features consider fragments of the image. This approach focuses on objects captured in the photograph, while disregarding the overall composition, which is partly dependent on the camera operator. Objects corresponding to uniform regions can be detected with the segmentation process described in *Datta et al. (2006)*. First the image is transformed in the LUV color space and the $K$ means algorithm is used to create $K$ color-based pixel cluster. Then a connected components analysis in an 8-connected neighborhood is performed to generating a list of all segments present. The 5 largest segments are denoted as $s_1, \ldots s_5$, in decreasing order. As most pictures contain many details resulting in noise, we applied a uniform blur with $m \times m$ ones matrix as kernel before the segmentation process.

*Rule of third:* A well-known paradigm in photography is that the main subject of attention in a picture should generally be in its central area. This rule is called the 'Rule of third' and the 'central area' can more precisely defined as the ninth of a photo divided by 1/3 and 2/3 of its height and width (see Figs. S2A and S2B). Using HSV color space $f_{52}$ defines the average hue $H$ for this region

$$f_{52} = \frac{1}{\left(\left[\frac{2M}{3}\right] - \left[\frac{M}{3}\right] + 1\right)\left(\left[\frac{2N}{3}\right] - \left[\frac{N}{3}\right] + 1\right)} \sum_{m=\left[\frac{M}{3}\right]}^{\left[\frac{2M}{3}\right]} \sum_{n=\left[\frac{N}{3}\right]}^{\left[\frac{2N}{3}\right]} I_H(m, n) \tag{39}$$

$I_S$ and $I_V$ are computed accordingly with $f_{53}$ and $f_{54}$.

*Focus region: Li & Chen (2009)*

offer a slightly different approach on the rule of thirds. The study suggests to use HSL color space and argue that focusing exclusively on the central ninth is too restrictive. From this approach, the focus region $FR$ was defined as the central ninth of the respective picture

plus a defined percentage $\mu$ in its immediate surrounding (Figs. S2A and S2B). For the here presented image analysis we set $\mu = 0.1$.

$$f_{55} = \frac{1}{\#\text{of}\{(m,n)|(m,n) \in FR\}} \sum_{(m,n) \in FR} I_{H\_}(m,n) \tag{40}$$

$I_{S\_}$ and $I_{L\_}$ are computed accordingly with $f_{56}$ and $f_{57}$.

*Segmentation:* The segmentation process generates a list $L$ of connected segments in which the 5 largest segments are denoted as $s_1, \ldots, s_5$. Our analysis focuses on the largest 3 or 5 segments only. Not only were the properties of each of these segments, but also the quantity of the connected segments in each picture recorded. This provides a proxy for the number of objects and the complexity of each recorded image.

$$f_{58} = \# \text{ of } L. \tag{41}$$

The number of segments $s_i$ in $L$ above a certain threshold ($f_{59}$), and the size of the 5 largest segments $s_i$ ($f_{60}$–$f_{64}$) was defined as:

$$f_{59} = \# \text{ of } \{s_i|\# \text{ of } s_i < MN/100, i \in \{1,\ldots,5\}\} \tag{42}$$

$$f_{59+i} = (\# \text{ of } s_i)/MN, \quad \forall \in \{1,\ldots,5\}. \tag{43}$$

To gain information on the position of these 5 biggest segments, the image was divided in 9 equal parts identical to Rule of third feature analysis. Setting $(r_i, c_i) \in \{1,2,3\}^2$ as the indices of the row and column around the centroid of $s_i$, features $f_{65}$ through $f_{69}$ as were defined, starting on the top left of each image as

$$f_{64+i} = 10 * r + c, \quad \forall \in \{1,\ldots,5\}. \tag{44}$$

The average hue, saturation and value were then assessed for each of the objects. Features $f_{70}$ through $f_{74}$ were computed as the average hues of each of the segments $s_i$, in the HSV color space:

$$f_{69+i} = \frac{1}{\# \text{ of } si} \sum_{(m,n) \in si} I_H(m,n), \quad \forall i \in \{1,\ldots,5\}. \tag{45}$$

Features $f_{75}$–$f_{79}$ and $f_{80}$–$f_{84}$ are computed analog for $I_S$ and $I_V$ respectively. Features $f_{85}$–$f_{87}$ were further defined as the average brightness of the top 3 segments:

$$f_{84+i} = \frac{1}{\# \text{ of } si} \sum_{(m,n) \in si} L(m,n), \quad \forall i \in \{1,2,3\} \tag{46}$$

lightness $L(m,n)$ has already been defined under 'Brightness analysis'. This allows us to compare the colors of each of the segments and to evaluate their diversity by measuring the *average color spread* $f_{88}$ of their hues. As complementary colors are aesthetically more pleasing together $f_{89}$ was defined as the *average complementary colors* among the assessed segments.
$$f_{88} = \sum_{i=1}^{5}\sum_{j=1}^{5} |h(i) - h(j)| \quad \text{and} \quad f_{89} = \sum_{i=1}^{5}\sum_{j=1}^{5} \|h(i) - h(j)\|_{al} \tag{47}$$

where $\forall_i \in 1,\ldots,5, h(i) = f_{69+i}$ is the average hue of $s_i$.

As round, regular and convex shapes are considered to be generally more beautiful, the presence of such shapes in a picture should increase its aesthetic value. Here we only assessed the shapes of the 3 largest segments in each image. The coordinates of the centers of mass (first-order moment), the variance (second-order centered moment) and skewness (third-order centered moment) was calculated for each of these segments was calculated by defining for all $i \in \{1,2,3\}$

$$f_{89+i} = \overline{x_i} = \frac{1}{\# \text{ of } si} \sum_{(m,n)\in si} x(m,n) \tag{48}$$

$$f_{92+i} = \overline{y_i} = \frac{1}{\# \text{ of } si} \sum_{(m,n)\in si} y(m,n) \tag{49}$$

$$f_{95+i} = \frac{1}{\# \text{ of } si} \sum_{(m,n)\in si} ((x(m,n) - \overline{x_i})^2 + ((y(m,n) - \overline{y_i})^2) \tag{50}$$

$$f_{98+i} = \frac{1}{\# \text{ of } si} \sum_{(m,n)\in si} ((x(m,n) - \overline{x_i})^3 + ((y(m,n) - -\overline{y_i})^3) \tag{51}$$

where $\forall (m,n), (x(m,n), y(m,n))$ are the normalized coordinates of pixel $(m,n)$.

Horizontal and vertical coordinates were normalized by height and width of the image to account for different image ratios. To quantify convex shapes in an image $f_{102}$ was defined as the percentage of image area covered by convex shapes. To reduce noise only R segments $p_1,\ldots,p_R$ containing more than $MN/200$ pixels were incorporated in this feature. The convex hull $g_k$ was then computed for each $p_k$. A perfectly convex shape $p_k \cap g_k = p_k$ and $\frac{\text{area}(pk)}{\text{area}(gk)} = 1$ would be too restrictive for our purposes of analyzing natural objects, so $p_k$ was considered convex if $\frac{\text{area}(pk)}{\text{area}(gk)} > \delta$.

$$f_{102} = \frac{1}{MN} \sum_{k=1}^{R} I\left(\frac{\text{area}(pk)}{\text{area}(gk)} > \delta\right) * |\text{area}(p_k)| \tag{52}$$

where $I(\cdot)$ is the indicator function and $\delta = 0.8$.

The last features using segmentation measure different types of contrast between the 5 largest segments. Features $f_{103}$–$f_{106}$ address the hue contrast, the saturation contrast, the brightness contrast, and the blur contrast. First the average hue, saturation, brightness, and the blur for each $s_i$ was calculated

$$h(i) = \frac{1}{\# \text{ of } si} \sum_{(m,n)\in si} I_H(m,n), \quad \forall i \in \{1,\ldots,5\} \tag{53}$$

$$s(i) = \frac{1}{\# \text{ of } si} \sum_{(m,n)\in si} I_S(m,n), \quad \forall i \in \{1,\ldots,5\} \tag{54}$$

$$l(i) = \frac{1}{\# \text{ of } si} \sum_{(m,n) \in si} L(m,n), \quad \forall i \in \{1, \ldots, 5\}. \tag{55}$$

To calculate the blur of the segment $s_i$, $I_{si}$ was computed so that

$$I_{Si}(m,n) = \begin{cases} (Ir(m,n) + Ir(m,n) + Ir(m,n))/3 & \text{if } (m,n) \in si \\ 0 & \text{otherwise} \end{cases} \tag{56}$$

and $b(i)$ defined as blur measure of $I_{si}$ for all $i \in \{1, \ldots, 5\}$, analog to the previously described 'Blur measure'.

$$b(i) = \max \left( 2 \frac{m' - \left[\frac{M}{2}\right]}{M}; 2 \frac{n' - \left[\frac{N}{2}\right]}{N} \right) \tag{57}$$

where $|Y_i(m', n')| > \theta, 0 < m' < M/2$ and $0 < n' < N/2$, with $Y_i = \text{fft } 2(I_{si})/\sqrt{MN}$ and $\theta = 0.45$. Features $f_{103}$–$f_{106}$ were then defined as

$$f_{103} = \max_{i,j \in \{1, \ldots, 5\}} (\| h(i) - h(j) \|_{al}) \tag{58}$$

$$f_{104} = \max_{i,j \in \{1, \ldots, 5\}} (\| s(i) - s(j) \|) \tag{59}$$

$$f_{105} = \max_{i,j \in \{1, \ldots, 5\}} (\| l(i) - l(j) \|) \tag{60}$$

$$f_{106} = \max_{i,j \in \{1, \ldots, 5\}} (\| b(i) - b(j) \|). \tag{61}$$

*Low depth of field indicators:* Finally, according to the method described by *Datta et al. (2006)* to detect low depth of field (DOF) and macro images, we divided the images into 16 rectangular blocks of identical size M1, …, M16, numbered in row-major order. Applying the notations of the 'Wavelet based texture', $C_3^H, C_3^V$, and $C_3^D$ denote the third level detail coefficient matrices generated by performing a three-level Haar wavelet transform on the hue channel of the image. The low DOF for the hue is then computed as

$$f_{107} = \frac{\sum_{(m,n) \in M6M7M10M11} (C_3^H(m,n) + C_3^V(m,n) + C_3^D(m,n))}{\sum_{i=1}^{16} \sum_{(m,n) \in Mi} (C_3^H(m,n) + C_3^V(m,n) + C_3^D(m,n))} \tag{62}$$

and $f_{108}$ and $f_{109}$ are calculated similarly for saturation and value.

## Machine learning

To reduce the noise and decrease the error, we analyzed multiple methods of determining feature importance. An unsupervised random forests approach was used to identify the most important features (Fig. S1). For every tree in the construction of a random forests, an out-of-bag sample was sent down the tree for calculation and the number of correct predictions was recorded. The variable importance was then generated by comparing the number of correct predictions from the out-of-bag sample to a randomly permuted

variant. For each feature, the resulting importance is:

$$\frac{1}{n_{\text{trees}}} \sum_{\text{all trees}} (R_{\text{OOB}} - R_{\text{perm}}).$$

A second method was to identify redundant columns before the training. Using a covariance matrix of the 109 features, relationships between columns were analyzed and columns with a correlation greater than 0.90 were clustered into groups. Within every group, features were either directly or mutually related. In order to not compromise the comprehensive approach of the coral reef aesthetic feature analysis the most important features from each group remained in the analysis while highly correlated, less important features within a group were removed. We built neural networks based on these two methods and discerned when removing redundant features we obtained lower mean square errors. Thus, we utilized a total of 97 features when building our ensemble of neural networks.

To fuse the predictive power of the 109 aesthetic features, a Levenberg–Marquardt algorithm was used simultaneously on every sample of the training set to minimize the mean squared error of the estimated output score and the NCEAS value. Typical mean squared error rates were in the 90s. We then decided on a threshold of 60 for the mean squared error and searched the weight space of the neural network to find 10 sets of weights with a mean squared error of less than 60 on the validation set. The predicted NCEAS scores of these 10 networks were then averaged for the ensemble prediction, which is our aesthetic value.

After running test data through the ensemble of neural networks, we further analyze the accuracy of our system by simultaneously testing multiple pictures at a time. To see how much more reliably we could deduce the NCEAS score using $N$ pictures from the same site, we averaged the outputs from our ensemble of neural networks for all twenty choose $N$ ($N = 1, 2, 3, 4, 5$) combinations available from the test batch. Combinations of multiple pictures increased the accuracy of the root mean square error of 6.57 for $N = 1$–5.35 for $N = 2$, 4.88 for $N = 3$, and 4.46 for both $N = 4$ and $N = 5$.

### Funding
The work was funded by the Gordon and Betty Moore Foundation, Investigator Award 3781 to FR. The funders had no role in study design, data collection and analysis, decision to publish, or preparation of the manuscript.

### Grant Disclosures
The following grant information was disclosed by the authors:
Gordon and Betty Moore Foundation: 3781.

### Competing Interests
The authors declare there are no competing interests.

## Author Contributions

- Andreas F. Haas conceived and designed the experiments, performed the experiments, analyzed the data, wrote the paper, prepared figures and/or tables, reviewed drafts of the paper.
- Marine Guibert, Sandi Calhoun and Emma George performed the experiments, analyzed the data, prepared figures and/or tables, reviewed drafts of the paper.
- Anja Foerschner conceived and designed the experiments, performed the experiments, analyzed the data, contributed reagents/materials/analysis tools, wrote the paper, reviewed drafts of the paper.
- Tim Co performed the experiments, analyzed the data, contributed reagents/materials/analysis tools, prepared figures and/or tables, reviewed drafts of the paper.
- Mark Hatay and Phillip Dustan performed the experiments, contributed reagents/materials/analysis tools, prepared figures and/or tables, reviewed drafts of the paper.
- Elizabeth Dinsdale conceived and designed the experiments, performed the experiments, contributed reagents/materials/analysis tools, wrote the paper, prepared figures and/or tables, reviewed drafts of the paper.
- Stuart A. Sandin and Jennifer E. Smith performed the experiments, contributed reagents/materials/analysis tools, reviewed drafts of the paper.
- Mark J.A. Vermeij performed the experiments, contributed reagents/materials/analysis tools, wrote the paper, reviewed drafts of the paper.
- Ben Felts performed the experiments, analyzed the data, contributed reagents/materials/analysis tools, reviewed drafts of the paper.
- Peter Salamon conceived and designed the experiments, performed the experiments, analyzed the data, contributed reagents/materials/analysis tools, wrote the paper, prepared figures and/or tables, reviewed drafts of the paper.
- Forest Rohwer conceived and designed the experiments, performed the experiments, analyzed the data, contributed reagents/materials/analysis tools, wrote the paper, reviewed drafts of the paper.

## Supplemental Information

Supplemental information for this article can be found online at http://dx.doi.org/10.7717/peerj.1390#supplemental-information.

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
