# Peer review of "Can we measure beauty? Computational evaluation of coral reef aesthetics"

_PeerJ, doi:10.7717/peerj.1390_

## Round 0.1 · original submission · Major Revisions

One of the reviewers had suggestions for major restructuring. The other had only minor suggestions. If you can revise the manuscript according to the reviewer's suggestions it would be acceptable for publication.

·

Basic reporting

I think the article could be improved with greater attention to structure (i.e. to the order in which ideas are presented).

For example, the article did not have headings for either the 'introduction' or 'materials and methods' although the discussion from lines 44 - 121 approximately followed that format. Discussion of methods during the beginning of the article was relatively brief (really only from lines 106-121) - although I note that after having presented results and having discussed those results, a much more detailed overview of methods is given (from lines 290 - 651). I presume this reflects the fact that the authors felt such detail was unnecessary to the main 'story'; and I must agree. However, it might make the article easier to follow if the detail were formally relegated to an appendix, and if the authors were to give a slightly expanded general overview of the methods in the main body of the paper prior to presenting results. This would make it much easier on readers when interpreting results (e.g. on line 142, reference is made to the 'training and test' set of images, but no mention had previously been made of 'training' and tests)

As regards the discussion, much of it seeks to justify the use of algorithms (or similar) to characterise aesthetic beauty - i.e. establishing that there is a link between colour and texture, and people's perceptions of beauty. That would be appropriate if this particular paper reported on the results of a study that assessed the ability of algorithms to characterise aesthetic beauty. But that does not seem to be what the study is doing. If I am understanding this paper correctly, it was a previous series of studies that established the link between people's perceptions of aesthetic beauty and computer scores. It is important to discuss those studies since that justifies the use of computer scores. But that justification does not belong in a discussion of (this paper's) results. Rather that discussion belongs in the methods section --- since it supports the choice of method (computer scores instead of surveys of perceptions).
What should the discussion focus on? The 'contribution' which this paper makes. I think this is primarily on the link between computer scores and biophysical indicators of the 'health' of the reef.

Experimental design

I quite like the paper and its general idea, which, if I understand it correctly is that
a) previous research has established that it is possible to characterise aesthetics/beauty using computer algorithms that describe various attributes of photographs (sorry, i know that oversimplifies, but I think it makes the point)
b) if one can establish a link between those computer generated 'aesthetic' scores and the biophysical condition of reefs, then it may be possible to use photographs (citizen science) to help monitor reef health.

But I think that some of that message is lost in structure - i.e. the order in which ideas are presented makes it difficult to follow that direct line of thought and to carefully distinguish between what has been done in previous work (a), and what is new or innovative about this paper ((b) and the policy implications of (a) and (b) together).

Validity of the findings

I do not have enough expertise to determine if the methods used to develop the computer scores for aesthetics are appropriate.

Assuming they are - and subject to my comments above about the desire to move some material from the 'discussion' into the methods section of the paper, it seems that the findings (of a link between computer scores for aesthetics and biophysical indicators of reef health) are both valid and interesting

Additional comments

As noted above. I like the topic, and think the findings have important implications - particularly for monitoring programs. But I think the paper needs to be restructured, to highlight those key contributions.

·

Basic reporting

Assessing ecosystem function/status can be an expensive and time consuming process. This paper demonstrates a non-invasive and effective way to assess coral reef status using photographs.

Experimental design

The technical details were outside my area of expertise.

Validity of the findings

As I said above, technical details regarding wavelet functions, machine learning, focal lengths, segmentation, etc. - Not my area of expertise. However, statistical correlation between the aesthetic values derived from the photographs and the microbial abundance measures look very positive and useful.

Additional comments

Very nice work. I loved how you characterized the variability of the photographs and microbial abundance and their covariation in Figure 3.

---

## Round 0.2 · accepted · Accept

Congratulations on an interesting and important paper.